# Impact of Remelting in the Microstructure and Corrosion Properties of the Ti6Al4V Fabricated by Selective Laser Melting

Javier Bedmar [ID], Jorge de la Pezuela, Ainhoa Riquelme [ID], Belén Torres and Joaquín Rams *[ID]

Department of Applied Mathematics, Materials Science and Engineering and Electronics Technology, Escuela Superior de Ciencias Experimentales y Tecnología (ESCET), Universidad Rey Juan Carlos, 28933 Mostoles, Spain; javier.bedmar@urjc.es (J.B.); j.delapezuela@alumnos.urjc.es (J.d.l.P.); ainhoa.riquelme.aguado@urjc.es (A.R.); belen.torres@urjc.es (B.T.)
* Correspondence: joaquin.rams@urjc.es

**Abstract:** The presence of defects like porosity and lack of fusion can negatively affect the properties of the materials manufactured by Selective Laser Melting (SLM). The optimization of the manufacturing conditions allows reducing the number of defects, but there is a limit for each manufacturing material and process. To expand the manufacturing envelope, a remelting after every layer of the SLM process has been used to manufacture Ti6Al4V alloy samples using an SLM with a $CO_2$ laser. The effect of this processing method on the microstructure, defects, hardness, and, especially, the corrosion properties was studied. It was concluded that the laser remelting strategy causes an increment of the α and β phases from the dissolution of metastable α'. This technique also provokes a decrease in the number of defects and a reduction of the hardness, which are also reduced with lower scanning speeds. On the other hand, all the corrosion tests show that a low scanning speed and the laser remelting strategy improve the corrosion resistance of the Ti6Al4V alloy since parameters like the Open Circuit Potential (OCP) and the Polarization Resistance (*Rp*) are nobler and the mass gain is lower.

**Keywords:** selective laser melting; $CO_2$ laser; Ti6Al4V; corrosion; laser remelting

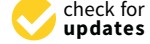



## 1. Introduction

Additive Manufacturing (AM) has become one of the most promising manufacturing techniques. AM processes are a set of techniques in which parts are fabricated layer by layer from an STL archive without the need for molds. One AM technique is the Laser Powder Bed Fusion (L-PBF) or Selective Laser Melting (SLM) which uses a laser to melt metal powder after depositing it in a powder bed in a protective atmosphere [1]. The SLM technique can fabricate near-net-shape pieces with complex and topologically optimized pieces without an increment of cost [2]. This allows the manufacturing of unique and personalized parts in different fields like jewelry [3], transport industry, aerospace field [4]; or biomedical sector [5].

However, the L-PBF technique has different negative qualities such as the long time that it takes to fabricate the parts or the number of different types of defects that the pieces can have. Several problems have been reported in studies like the residual stress [6] or the anisotropy [7], but those that are the most important are porosity [8] and lack of fusion [9]. While porosity is related to gas entrapped during the solidification or the sublimation of some alloy elements, the lack of fusion is linked to a low amount of energy during the melting of the powder [10]. These defects are critical of the use of the SLM samples in the aerospace field because they have a strong impact on their toughness and fatigue resistance.

To reduce the number of these defects, different solutions have been proposed. The first one is the appropriate selection of the manufacturing parameters. For it, an adequate combination of laser power, scanning speed, and hatch distance is critical. Using lower laser power of very high scanning speeds causes lacks of fusion, as the irradiated powder is not fully molten, leaving unmolten particles within the microstructure of the material.

On the other hand, using higher laser powers and lower scanning speeds increase the amount of molten material, not only of the powder in the bed but also of the previously molten material. In these conditions, a relevant part of the material is subjected to multiple melting during the fabrication of different layers. Furthermore, the higher input heat per unit volume makes the local temperature higher. Therefore, cooling is slower, providing more time for the evolution of the microstructure than with lower energetic conditions. The hatch distance has also a relevant effect on the proportion of remolten material, which increases as the hatch distance decreases, and causes a reduction in hardness.

Different types of lasers have been used in L-PBF, the most used one being fiber lasers (FL). These lasers are very stable and versatile, have a high quantum efficiency, and provide good quality samples due to their wavelength, near to the infrared, which provides a high absorbance for metals. On the other hand, they have some polarization instabilities and non-linear effects in the gain medium which can limit its performance [11]. An alternative technique is the use of $CO_2$ lasers, which have found an increase in their use in the last years. In these lasers, a gas is electrically pumped by a current (DC or AC) to induce the population inversion required for laser emission. $CO_2$ lasers provide a high efficiency (20%), output power from 0.1 to 20 kW, and are low-cost systems that are relatively simple and reliable. Due to this, $CO_2$ lasers are used in different applications: cutting, welding, marking, surface modification, etc. The main disadvantage of these lasers is that they emit light at a 10.6 μm wavelength, which is very difficult to be controlled. This results in beams with greater divergence and a greater laser focus size [12]. They also have a low absorptivity by many metals, such as aluminum [11]. Despite all the disadvantages of these lasers, they are available in a wide range of powers, beam qualities, and laser modes, so it is necessary to explore the limits of this technology to apply it in SLM equipment to investigate the possibility of implanting low-cost 3D printers for metals.

Another alternative that is currently being used is the application of postprocessing, like heat treatments [13]. However, some authors, such as Karimi et al. [14], have investigated, in titanium, the influence of the remelting of every layer during the manufacturing process. These studies have concluded that defects such as porosity and lack of fusion are reduced because of the repeated melting of every layer. However, no studies have been found about the effects of these additional meltings on the corrosion behavior.

Ti6Al4V alloy is one of the most commonly used in the 3D printing field due to its low distortion compared to the one reached by conventional methods [15], the fine microstructure it has when a laser melts it [16] and its good weldability if the conditions are controlled and optimized [17]. On the other hand, the lightness, high mechanical properties, and corrosion resistance distinguish this alloy [18]. These properties make this alloy very useful in several areas, like the aerospace [19] or the energy industries [20], and the prosthesis field [21]. However, Ti6Al4V pieces manufactured by L-PBF are not used in the aerospace area because the defects that appear limit their fatigue and corrosion resistance. Any improvement in their manufacturing would result in widening the application of SLM samples in the transportation field.

L-PFB Ti6Al4V shows several differences from the same alloys manufactured by conventional processes, due to the layer-by-layer deposition and the high cooling speed during the solidification of the material [22]. This is the cause of the different properties of the Ti6Al4V depending on its fabrication method. Parts made by L-PBF are harder than the manufactured by conventional processes [23] but they show poorer wear resistance [24]. In general, the properties of L-PBF are close to those of wrought structures. In both systems, the mechanical properties are associated with the platelet structure of the alfa + beta phases, which upon cooling transform into a dendritic beta microstructure, which depends on the cooling process. On the other hand, several studies have demonstrated that the corrosion resistance of these AM alloys is superior [25] in the case of the parts fabricated by SLM equipped with a fiber laser. However, there are no studies in the case of the SLM with $CO_2$ laser.

In this study, Ti6Al4V parts have been manufactured by SLM using a $CO_2$ laser, and its microstructure, hardness, and corrosion properties have been studied. Furthermore, to improve the corrosion properties, a laser remelting of each layer during the manufacturing process has been studied. The corrosion properties obtained have been characterized and associated with changes in the microstructure of the samples and have been compared to parts fabricated by fiber laser.

## 2. Materials and Methods

### 2.1. Materials Manufacturing

Ti6Al4V powder, supplied by Renishaw was used to fabricate parts by L-PBF. The powder had a nominal composition (in % wt) of Aluminum (5.50–6.50), Vanadium (3.50–4.50), Iron (<0.25), Oxygen (<0.13), Carbon (<0.08), Nitrogen (<0.05), Hydrogen (<0.012), Yttrium (<0.005), residuals (<0.4), and Titanium (rest). The powder size was below 40 μm and had a spherical shape (Figure 1a) with a particle size distribution shown in Figure 1b.

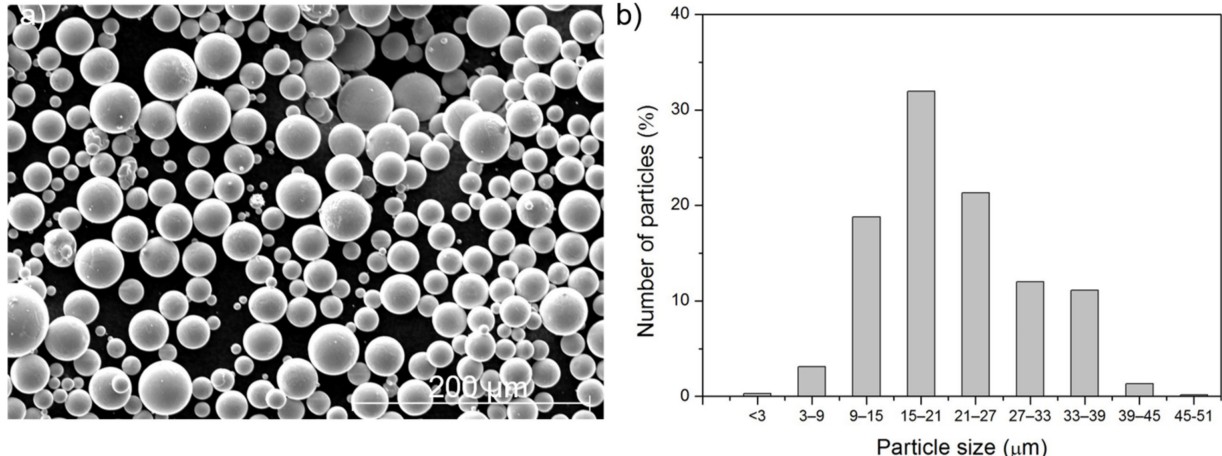

**Figure 1.** (**a**) SEM image of the Ti-6Al-4V powder; (**b**) particle size distribution of the powder used.

The SLM equipment (Aurora Labs, Canning Vale, Australia) was an S-Titanium Pro L-PBF system, which has two $CO_2$ lasers with a power of 150 W each, which was used in previous studies from different universities with good results [26,27]. The manufactured specimens were built with the orientation shown in Figure 2a and had dimensions of 20 mm × 20 mm × 3 mm, Figure 2b. During the fabrication, the build plate and the powder were preheated to 70 °C. The scanning strategy had a rotation of 67° between layer and layer, the hatch distance was 100 μm and the layer height was 60 μm. The 67° angle was chosen as it is the one used in many industrial L-PBF systems and appears in most of the studies on this topic [28] because it avoids the accumulation of parallel scans between closer layers, resulting in a more isotropic structure [29]. An Argon atmosphere was used to maintain, during all the manufacturing process, an oxygen level below 0.4%.

The samples were manufactured using 300 W of laser power, using the combined power of two lasers of 150 W that were focused in the same point; and the scanning speeds used were 50, 33, and 20 mm/s. A set of samples was manufactured using the standard methodology of melting the metallic powder after its deposition with the scanning speeds indicated.

A second set was manufactured using a double laser scanning method. Firstly, a layer of powder is spread on the surface. Then, the laser scans the programmed zones and there melts the powder. Afterward, after finishing the first scan in all the programmed zones, the laser repeats the previous pattern and applies a second melting process, without adding more powder. From the first scan to the second one, the elapsed time was 10 s. Then, the next powder layer was deposited, and a 67° rotation angle between the laser scanning was applied. Table 1 resumes the different types of samples manufactured. The samples with

a scanning speed of 20 mm/s could not be made manufactured with a second set due to excessive heat, which caused its partial melting. The heat input (W/m) in Table 1 has been calculated as the input laser power (W) divided by the scanning speed (mm/s).

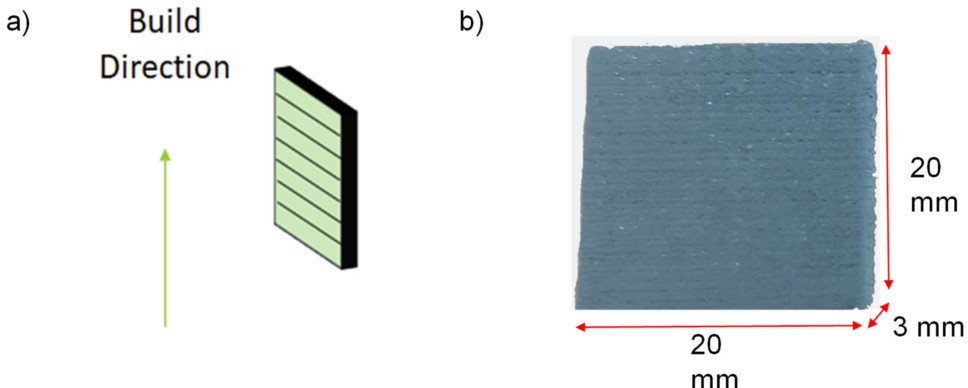

**Figure 2.** (**a**) Scheme of the build orientation of the manufacture samples and (**b**) a Ti6Al4V manufactured sample.

**Table 1.** Manufacturing conditions used in the manufactured samples.

| Sample | SLM Scanning Speed (mm/s) | Heat Input (W/mm) | Remelting |
|---|---|---|---|
| Ti-20 × 1 | 20 | 15.0 | No |
| Ti-33 × 1 | 33 | 9.1 | No |
| Ti-33 × 2 | 33 | 9.1 per scan | Yes |
| Ti-50 × 1 | 50 | 6.0 | No |
| Ti-50 × 2 | 50 | 6.0 per scan | Yes |

*2.2. Microstructural Characterization*

The microstructural characterization of the fabricated specimens was carried out by Optical Microscopy (OM) using a Leica DMR equipped with Leica Image Pro plus software and by Scanning Electron Microscopy (SEM) using a Hitachi S-3400N microscope equipped with an Energy Dispersive X-Ray Spectrometer (EDS, Bruker AXS Xflash Detector 5010, Bruker AXS Microanalysis, Berlin, Germany). The samples were prepared by polishing and etching them after embedding them. The etch used was Kroll's reagent (50 mL of distilled water, 25 mL of $HNO_3$, and 3 mL of HF). Furthermore, X-ray diffraction (XRD, Philips Analytical Company, Eindhoven, The Netherlands) using a Philips X'Pert diffractometer (CuKα = 1.54056 Å) was used to identify the phases formed in the samples.

Microhardness tests were performed by using a Microhardness Tester (HMV-2TE, Shimadzu, Kioto, Japan). The applied loads were 980.7 mN (HV0.1) for 15 s on the polished samples with a Vickers indenter. The average hardness was calculated from a minimum of fifteen indentations.

*2.3. Corrosion Tests*

Electrochemical tests were carried out using a three-electrode cell configuration: the sample acting as the working electrode, a silver/silver chloride (Ag/AgCl, KCl 3M) as a reference electrode, and a graphite rod as counter-electrode. The tests were performed in a 3.5% NaCl aqueous solution at room temperature with an Autolab PGStat302N potentiostat (Metrohm AG, Herisau, Switzerland). Before testing the materials, they were ground up to 1200 grit.

Lineal polarization tests were carried out by varying the potential ±10 mV around the open circuit potential (OCP) with a scanning rate of 1 mV/s. The polarization resistance (*Rp*) was derived from the corrosion current density and the applied potential using Ohm's

law. The samples were immersed in the chloride solution for up to one week time and the tests were made in triplicate at the following immersion times: 1, 6, 24, 48, 72, 96, and 168 h.

Anodic–cathodic polarization measurements (Tafel tests) were performed by polarizing the specimens between −800 mV and 1 V around the corrosion potential ($E_{corr}$) with a scanning rate of 1 mV/s. To get stable corrosion measurements, they were made after 1 h of immersion time. Furthermore, cyclic polarization tests were conducted to determine the presence of pitting corrosion. The curves were obtained by anodically polarizing the sample until the pitting potential ($E_{pit}$) was reached, and then the potential was reduced until $E_{corr}$ was reached again. In practice, the samples were polarized between −1.4 V and 1 V and again to −1.4 V with a scanning rate of 0.1 V/s and a return speed of 0.005 V/m after 1 h of immersion in the NaCl solution.

The mass gain of every specimen was measured for up to a week of immersion in the 3.5% NaCl aqueous solution to evaluate the progression of corrosion in the different samples manufactured.

After the immersion tests, the samples were observed by OM and SEM to identify the progress of corrosion in the samples.

## 3. Results and Discussion

### 3.1. Microstructure and Defects

The cross-section of the samples manufactured using the methodology indicated in the experimental section with only one melting per layer can be observed in Figure 3. The sample manufactured with a scanning speed of 20 mm/s, Figure 3a, was the one that received the highest laser input energy. It showed few defects, which consisted of voids and lacks of fusion, mainly located between the deposited layers. The samples manufactured at 33 mm/s (Figure 3b) and 50 mm/s (Figure 3c) showed similar features although the number of defects increased with increasing scanning speed, i.e., reducing the input heat. This indicates that using higher input energies reduces the presence of defects, particularly the lack of melting. However, in some places where the material was molten, there existed some porosity. This could be explained by possible oxidation of the surface of the samples. In this case, the use of higher laser power would not reduce further the porosity. Furthermore, using laser scanning speeds below 20 mm/s was not possible as it caused the degradation of the samples.

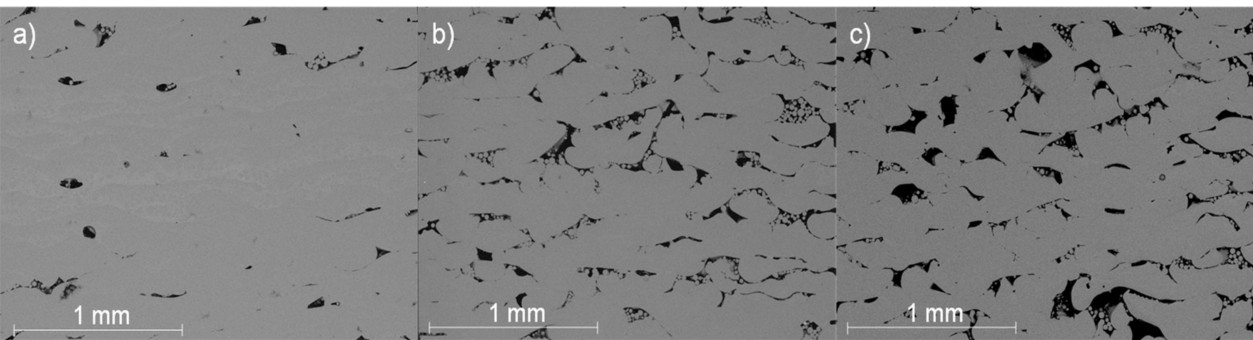

**Figure 3.** Optical images of the samples: (**a**) Ti-20 × 1; (**b**) Ti-33 × 1; (**c**) Ti-50 × 1.

To reduce further the porosity, the samples were scanned twice. In the first one, the powder was deposited and molten, and in the second one, the material was submitted to a second fusion with the sample power used for the first one. This second scan reduces the porosity, so the thermal conductivity of the material improves, having a higher cooling rate for the melted layers and a reheating for the layers below.

The sample manufactured at 20 mm was fully molten when applying a second step at 20 mm/s and was rejected from the study. Figure 4 shows the cross-sections of the samples fabricated with this strategy using scanning speeds of (a) 33 mm/s, and (b) 50 mm/s. It

can be appreciated that the number and size of the pores reduced, indicating that (i) the material can flow when it is heated again, and (ii) that the presence of oxygen at the surface is not predominant as the molten material can wet the previously deposited layer.

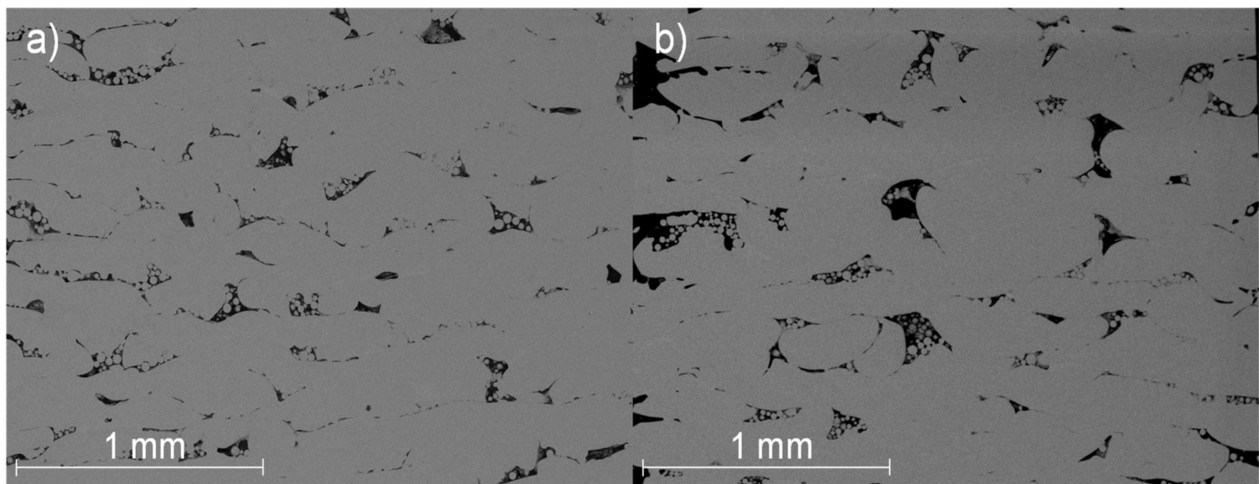

**Figure 4.** Optical images of the samples: (**a**) Ti-33 × 2; (**b**) Ti-50 × 2.

The second laser scan affected the kinds of defects present in the samples. In SLM samples, two kinds of defects are frequently observed: pores and lack of fusion. On the one hand, pores are associated with the presence of trapped gas in molten metal that gives rise to spherical defects (black arrows in Figure 5). On the other hand, lack of fusion is caused by insufficient energy during laser melting and the defects have concave zones as the unmolten particles retain part of their shape (white arrows in Figure 5).

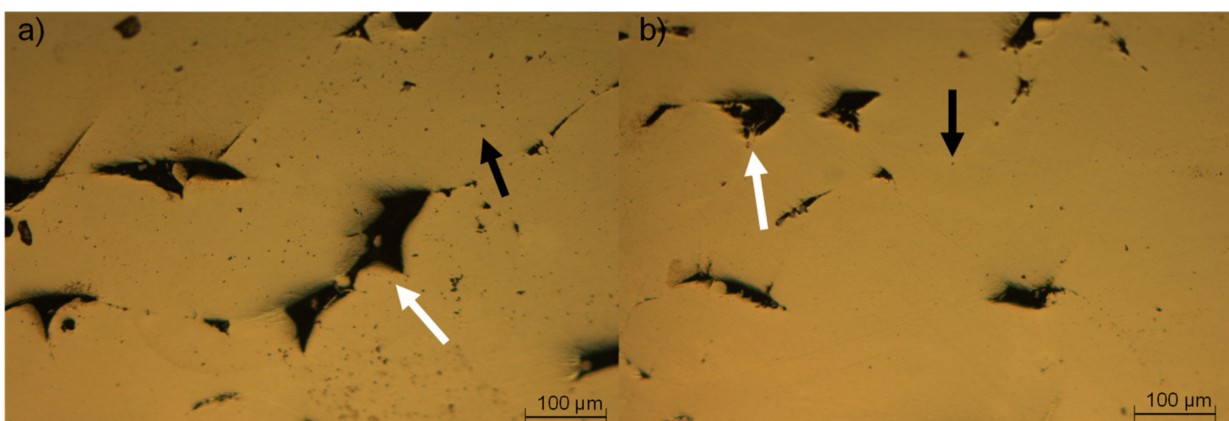

**Figure 5.** Porosity (arrowed in black) and lack of fusion (arrowed in white) defects on the manufactured samples: SEM micrograph of (**a**) Ti-50 × 1 and (**b**) Ti-50 × 2 samples.

In the samples manufactured using the standard procedure, the number of both types of defects was reduced by increasing the laser power used or by reducing the laser scanning speed, as was shown in Figure 3. However, in L-PBF with $CO_2$ laser it was not possible to eliminate most of the porosity as the parts fully melt before closing the defects. By applying the double laser melting treatment, both kinds of defects were reduced. The most evident change affected the melting faults, which reduced as the layer material was remelted and could flow to close the defects. In the case of porosity, by remelting the powder layer, more time was allowed for the gas trapped inside the material to be released. In both cases,

defects were reduced, and the densification of the samples was increased compared to as-built samples.

This evolution has a significant change in porosity, i.e., in relative density, as is shown in Figure 6. The porosity values measured were in the 4–11% range, which is greater than those of samples manufactured using other SLM equipment. In our case, the energy source is a $CO_2$ laser, which has an intrinsic instability associated with using a gas as the source of energy. A result of the thermal evolution of the gases causes turbulences in the active medium, which causes distortions in the beam intensity and shape, and also in the polarization of the emitted light. Furthermore, the $CO_2$ laser beams propagate in air and are directed to the samples by using mirrors from the lasers to the substrates. The travel of the laser light through the air also causes distortions in the beam. Finally, in SLM the laser must be turned on and off, at least once every layer, and on many occasions more, so the gas does not reach a steady state.

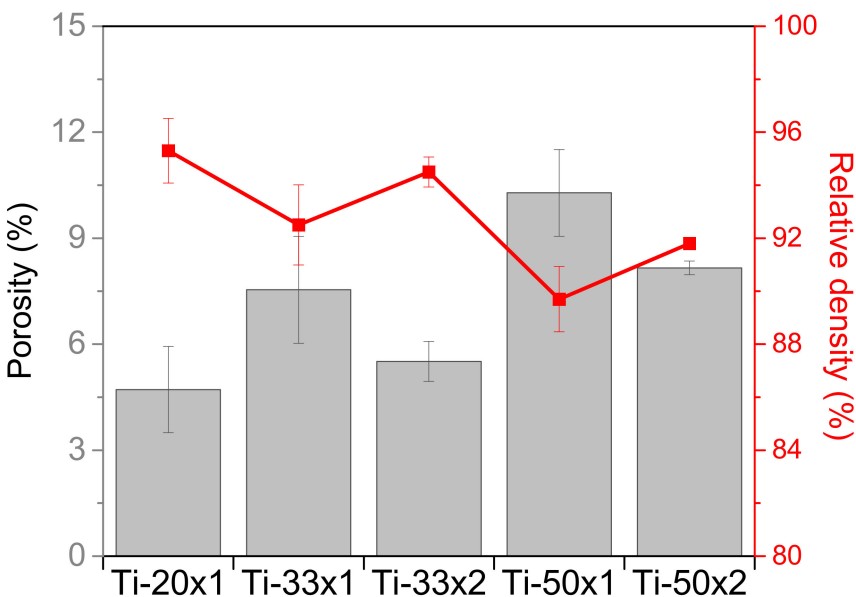

**Figure 6.** Porosity and relative density of the manufactured samples.

Apart from this, differences were observed between the samples. Porosity in samples made by one laser scan per layer is reduced as the laser scanning speed reduces. For the conditions used, porosity was reduced by 35% by using 33 mm/s instead of 50 mm/s and reduced further by another 40% by reducing it to 20 mm/s. Therefore, the manufacturing condition with the lowest porosity was the Ti-20x1. The increase of the laser input heat from 6.0 W/mm to 9.1 W/mm raised the relative density from 89.7% to 92.5% and increasing further the heat input to 15.0 W/mm improved further the relative density to 95.3%. The application of a second laser pass of 6.0 W/mm increased the relative density to 91.8% and in the case of the use of the second pass of 9.1 W/mm increased to 94.5%. Therefore, the higher the heat input, the higher the relative density of the samples. However, an excess in the input heat could melt the samples. Therefore, heat input values above 15 W/mm in the first scan melt the samples, and values above 9.1 W/mm melt the samples in the second scan, with the manufacturing conditions used in this work.

The influence of the double laser scan was relevant. In the case of the double scan in the pieces made at 50 mm/s, porosity was reduced by 20%, while in the case of the pieces made at 33 mm/s the reduction of the porosity was 30%. After the two laser scans, the porosity of the Ti-33 × 2 samples was near to that of the Ti-20 × 1 specimen. As indicated before, two laser scanning was not made at 20 mm/s because the sample melted and could not be fabricated. In any case, the porosity of the samples fabricated was above that of other studies that used fiber lasers [30,31].

Figure 7 shows the XRD pattern obtained for the manufactured samples with details of the β-phase peaks. Most of the peaks found in the spectra correspond to the α-phase, which is predominant in the alloy used and is not dependent on the manufacturing conditions used. On the other hand, it can be observed that the β-phase has a residual presence as their peaks, which appear at 39°, 63.5°, and 87°, are much smaller. Furthermore, the peak at the shorter angle is overlapped by the α peaks and cannot be distinguished. Finally, it must be noted that the α peaks coincide with the Widmanstatten α peaks, so the formation of this phase must not be excluded.

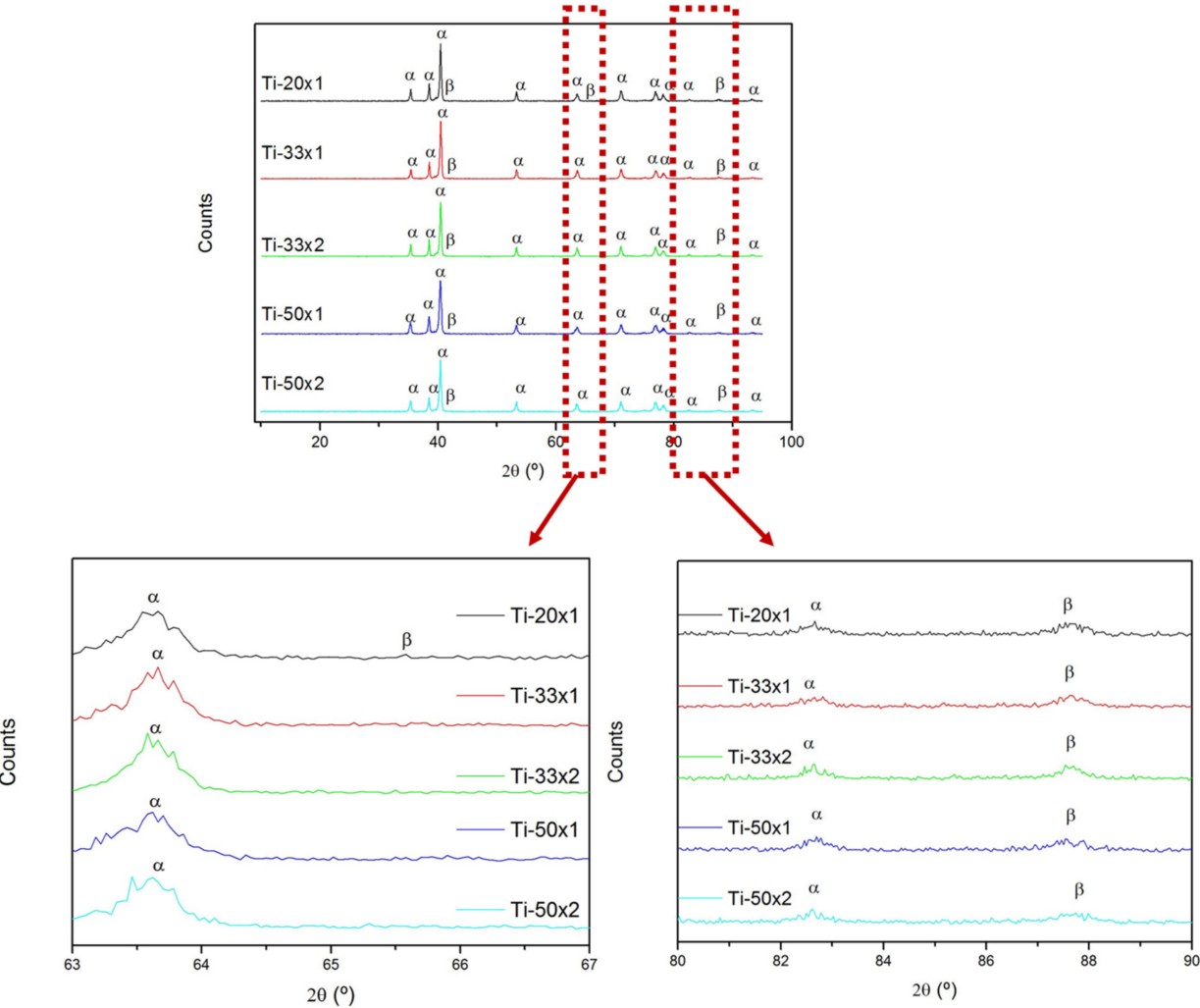

**Figure 7.** XRD spectra of all samples with details of the peak at 63.5° and 87.

The spectra obtained for the samples with or without the remelting of its layers are very similar, but slight differences in the ratio between the α and β peaks were observed. All samples showed a peak at 87° that corresponds to the β phase, and the Ti-20 × 1 sample also showed a peak at 65.7°, which slightly arose from the noise and corresponds to the (200) plane of the β phase [32].

It was observed that the phase that is more prone to corrosion is the β one, so the greater the presence of the β phase the greater the corrosion susceptibility of the samples. The proportion of the β phase can be correlated with the proportion of the diffraction peaks of the α and β phases, and the higher the α/β ratio, the better the corrosion behavior of the samples [33]. According to this criterion, the sample that could be more prone to corrosion is the Ti-20 × 1 sample. This shows that reducing the scanning speed to reduce the presence of pores and defects may have a detrimental effect on the phases formed in

the microstructure. In the samples manufactured at 33 and 50 mm/s of scanning speed, the effect of remelting in the presence of the β phase was dissimilar as it increased in the former and reduced in the latter one.

Three different phases were observed by optical microscopy in every specimen (Figure 8a): acicular martensitic α and α′ which appeared combined in all the zones, and β phase at the grain boundaries (GBs). β phase appeared during the solidification of the molten pool as grains that grow along the heat dissipation direction [34] (Figure 8a). Since this direction varies because of the different scanning directions and the successive remelting, these grains developed in different orientations. As the cooling continued, the β phase transformed into α and α′ phases, which, since the cooling is fast, appeared as needles due to a martensitic transformation (Figure 8b). The distinction between α and α′ is that, due to fast cooling rates, the Vanadium of the alloy cannot segregate along the molten pool, this makes needles of α with high concentrations of Vanadium which are the final α′ phase [23]. However, the α and α′ needles cannot be distinguished by XRD, optical microscopy, or SEM because they have the same crystalline structure and morphology. The β → α transformation made that the β phase remained only at the border of the prior β grains (Figure 8b).

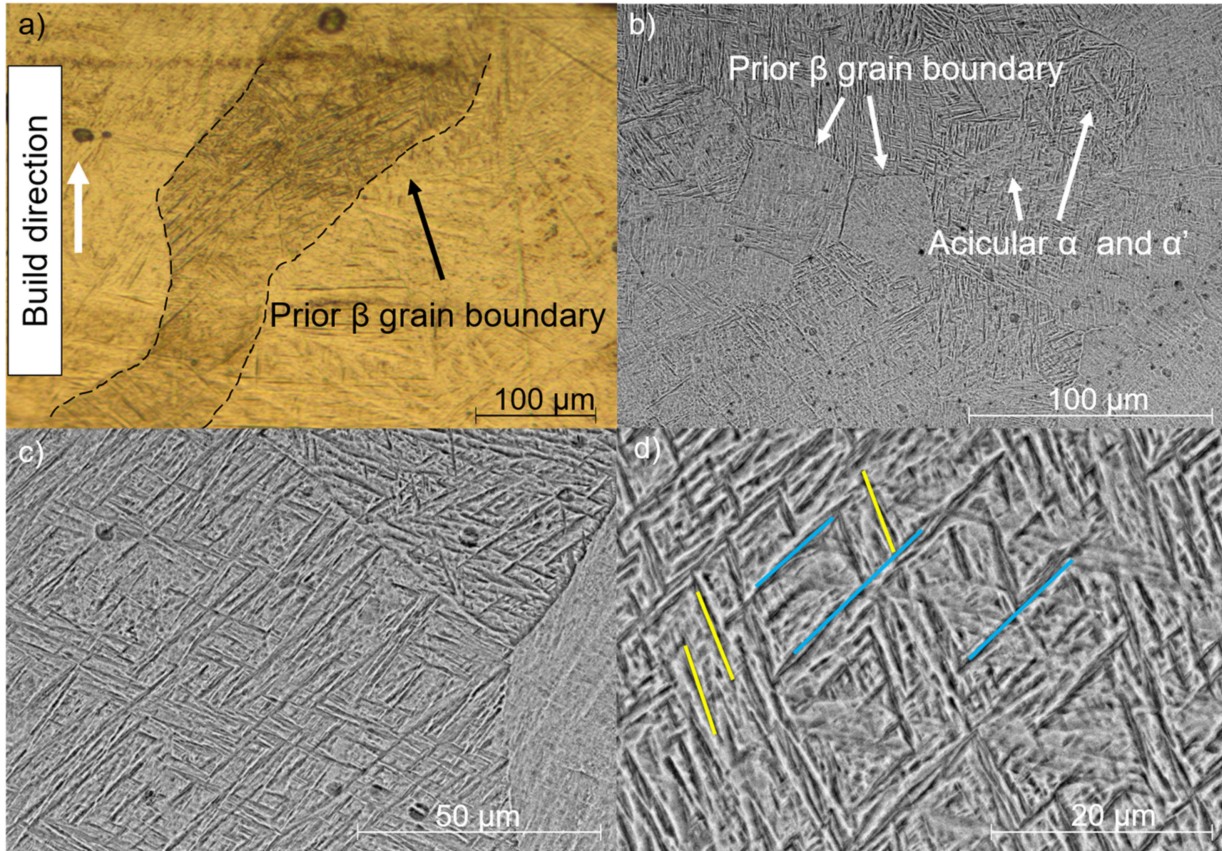

**Figure 8.** Microstructure of the Ti-33 × 1 specimen: (**a**) OM; (**b**) SEM; (**c**) detail of b; and (**d**) detail of the orientation of the needles formed, which are parallel in different directions (yellow for one direction and blue for another).

Finally, due to the consecutive laser scans of the process, the printed zones suffered some reheating which could promote the segregation of the Vanadium making α′ transform into β and α phases [32]. Although the α and α′ needles appear in random directions (Figure 8c), they are parallel between them (Figure 8d), and differences between them are associated with the texture of the sample and not on the inner morphology of each grain.

The grain texture was studied by SEM and differences in the grain size and distribution were observed. The grain size of the Ti-20 × 2 (Figure 9a) was larger than that of the Ti-33 × 1 (Figure 9b) and both greater than that of the Ti-50 × 1 (Figure 9c). Therefore, the grain size reduces when increasing the laser scanning speed, i.e., reducing the heat input. The grains found in the Ti-33 × 2 (Figure 9d) and Ti-50 × 2 (Figure 9e) were greater than the samples with only one laser pass. Therefore, the application of the second scan increased the grains size. Apart from the changes in the number of defects previously described.

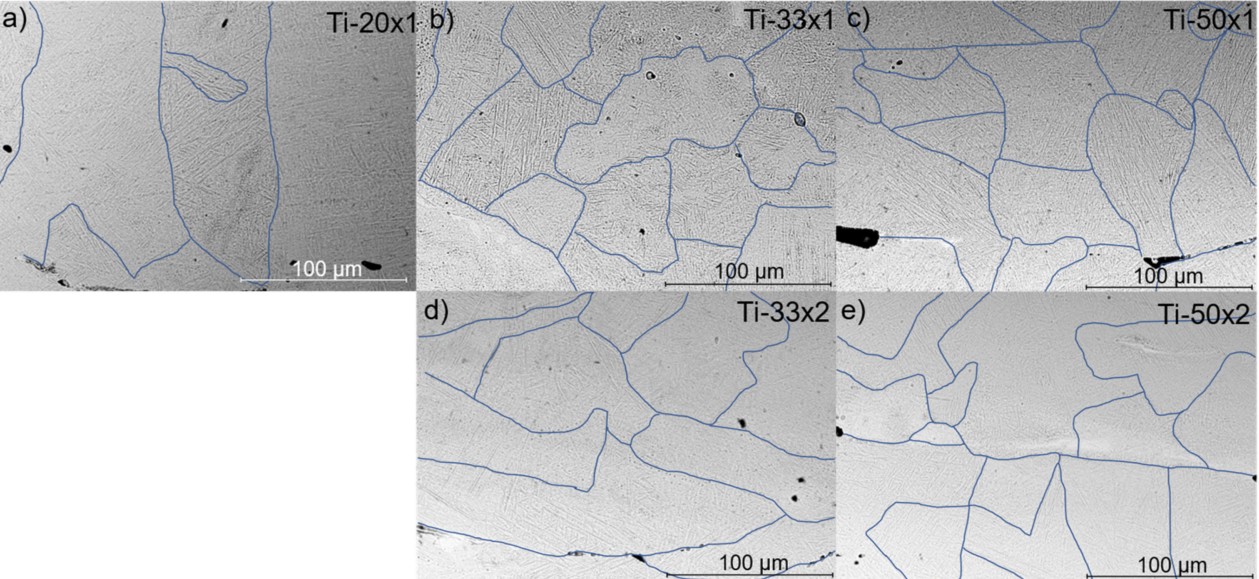

**Figure 9.** Microstructure of the different specimens tested as observed by SEM: (**a**) Ti-20 × 1; (**b**) Ti-33 × 1; (**c**) Ti-33 × 2; (**d**) Ti-50 × 1; and (**e**) Ti-50 × 2.

*3.2. Microhardness*

The results of the microhardness tests made to the samples are shown in Figure 10. In all cases, the hardness was above the 408 $HV_{0.2}$ reported for the same alloy manufactured by L-PBF with a fiber laser [35]. These values are also greater than the 342 $HV_{0.1}$ obtained by casting, and the 360 $HV_{0.1}$ by Hot Pressing [36], All the samples had a hardness 16% higher than the cast alloy, and the value for the Ti-33 × 1 sample was 30% above it.

For the one laser-scanned samples, the hardness increased when reducing the speed from 50 mm/s to 33 mm/s but decreased if the laser speed was reduced to 20 mm/s. This can be explained by the combined effect of the reduction in the porosity associated with the use of slower laser scans, but when the heat provided to the samples is higher, the cooling speed reduces. This causes increases in the grain size and results in a reduction in the hardness of the samples.

In all cases, the application of a second laser scan caused a slight reduction in the microhardness of the samples. The remelting process has two opposite effects on hardness [37]. On the one hand, we observed that the application of multiple remelting and reheating processes promoted the growth of the grain size, and possible the transformation of the α′ phase into the α and β phases. Both effects could cause a reduction of the microhardness of the samples. On the other hand, the increase in the relative density of the samples could increase hardness and thermal conductivity. For the conditions tested in this study, the result of these changes was a slight reduction in hardness. However, another study indicated that hardness increased with the application of one or two remelting processes using a fiber laser [14]. The main difference between the works is the input heat, which was 0.6 W/mm in their case and higher than 6 W/mm in ours. The order of magnitude in difference indicates that, at all conditions, the remelting we caused was much greater and,

hence, the cooling rate was slower. They showed this phenomenon by a reduction in the lattice parameters, although they did not study the grain size.

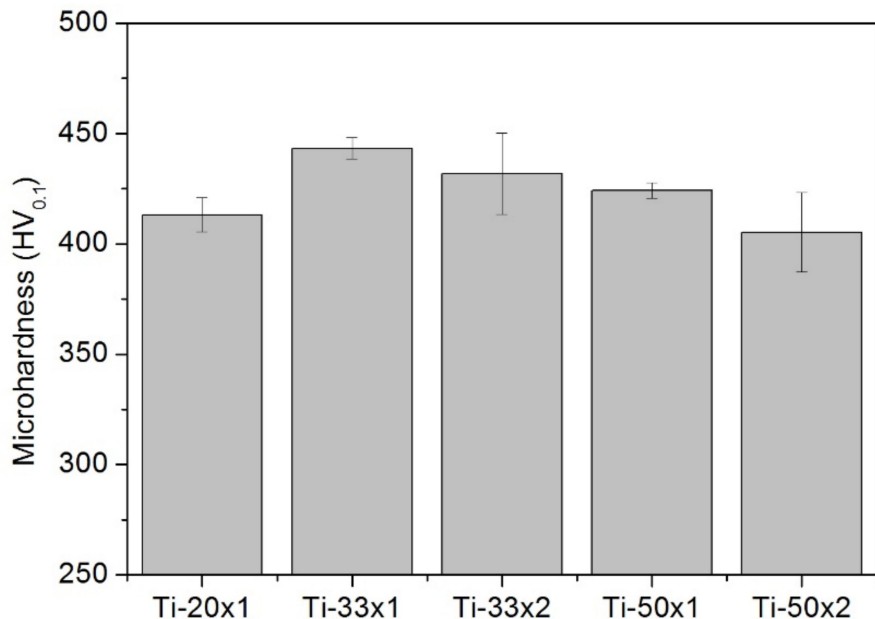

**Figure 10.** HV$_{0.1}$ microhardness of the samples manufactured.

*3.3. Corrosion Behavior*

3.3.1. Linear Polarization Resistance (LPR)

Figure 11a shows the results of the OCP tests measured for up to a week on immersion time. The OCP values were stable during the test and were very similar to each other. The only sample that showed a change in their properties was the Ti-33 × 1 samples, which ennobled for the first hours and then remained constant for the rest of the test.

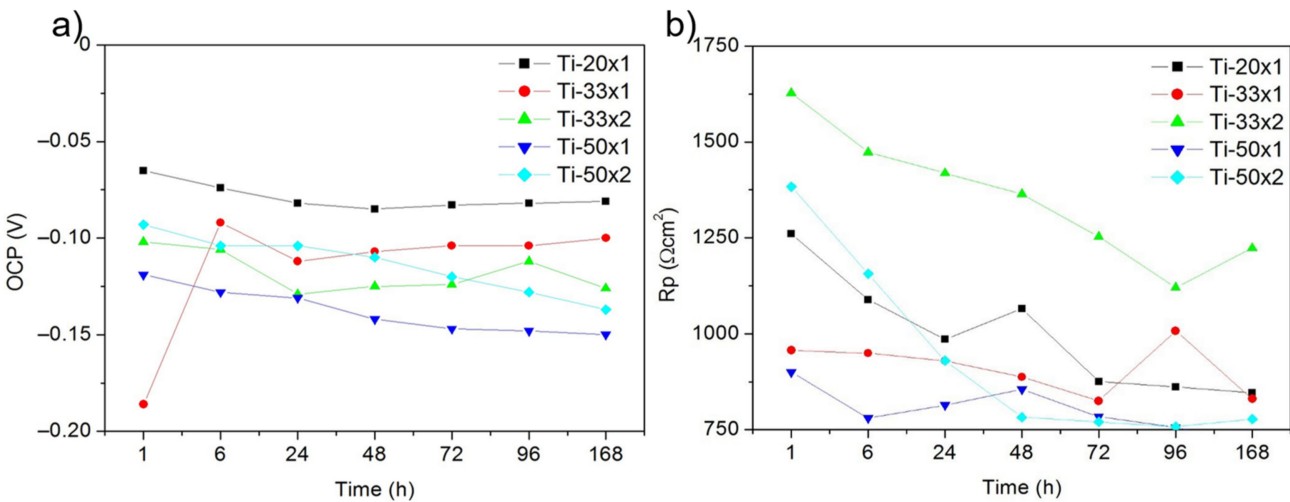

**Figure 11.** OCP of each specimen measured along a week (**a**); and *Rp* of each specimen measured during a week of immersion in 3.5% NaCl (**b**).

The values were nobler, i.e., with higher OCP value, when manufactured at lower scanning speed. Within the samples manufactured by two laser scanning, the Ti-50 × 2 was nobler than the Ti-50 × 1, while the Ti-33 × 2 was nobler than the Ti-33 × 1 only in the first hours and at 6 h of immersion time was less noble.

It was observed that, in the first hours, the OCP values of the samples made by $CO_2$ laser were better than the OCP values reported for parts made by fiber laser with a heat input of 0.16 W/mm, which were below −0.20 V the first 15 h [38]. Although in the subsequent hours, the samples made by fiber laser became nobler and the OCP of the specimens made by $CO_2$ laser was much more stable with time.

Figure 11b shows the results of the *Rp* of the as-built and remelted samples. In general, a low scanning speed improved the *Rp* parameter. The Ti-20 × 1 had greater polarization resistance, due to its lower porosity than the other as-built samples. Furthermore, the application of a double laser pass increased the *Rp* values, multiplying them by 2. This implies a significant reduction of the corrosion susceptibility of the samples.

The *Rp* of all the samples decreased with time. In the case of those with the double laser pass, the best performance was observed for the Ti-33 × 2 sample, while the Ti-50 × 2 had very high *Rp* values for the first immersion hours, but they reduced with time. However, the *Rp* values of the Ti-50 × 2 were above those of the as-built sample manufactured at the same speeds. The increase of the polarization resistance in the samples with the double laser pass was due to several factors: lower porosity [39], reduction of the number of metastable phases [33], and diminution of the residual stresses [40].

It is relevant the *Rp* of the Ti-33 × 2 specimen, which is the highest along the week, higher even than the Ti-20 × 1 one. It can be concluded that laser remelting had a big influence on the intermediate scanning speed, showing that the corrosion behavior achieved is better by remelting every layer than by reducing the scanning speed.

### 3.3.2. Tafel Tests

Scanning electrochemical tests (Figure 12) showed that lower scanning speeds improve corrosion behavior. This was confirmed by a lower current density and a higher corrosion potential and is caused by the lower densification of the material. In parts made by SLM with fiber laser by other authors with a heat input from 0.23 to 0.46 W/m, the $E_{corr}$ was below −0.55 V [39] and did not reach the values of the parts made in this work by SLM with $CO_2$ laser.

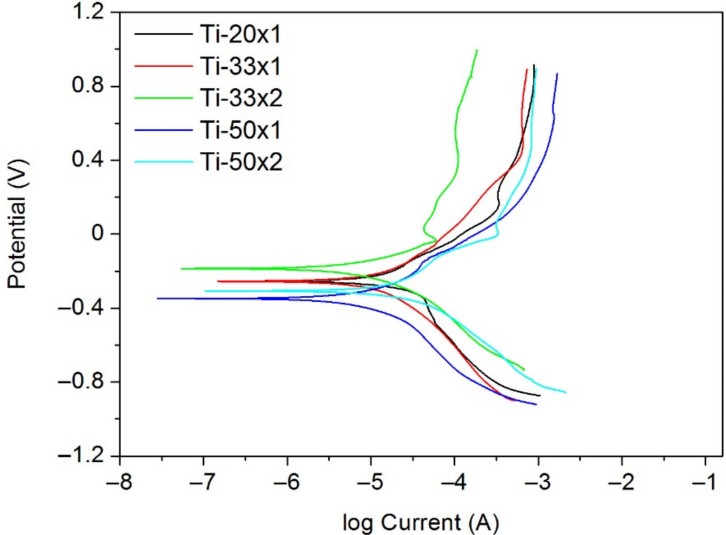

**Figure 12.** Tafel tests after one hour of immersion in 3.5% NaCl.

It can be affirmed that the behavior of the samples with double laser (Ti-33 × 2 and Ti-50 × 2) was better than that of the as-built samples. For the Ti-33 × 2 sample, the improvement was greater than for the samples manufactured at 50 mm/s. On the other hand, as it was shown in the previous section, within the working range of this study, for intermediate scanning speeds, 33 mm/s, the laser remelting treatment resulted in a better

corrosion behavior than a reduction of the scanning speed, but this does not occur for the high scanning speed tested, i.e., 50 mm/s.

In all cases, with one and two meltings, the parts made by $CO_2$ laser reached better results than those shown by other parts fabricated by SLM with fiber laser (as-built and with postprocessing) in similar solutions [25], showing that SLM by with $CO_2$ laser is competitive with other techniques for the manufacturing of Ti6Al4V parts from the corrosion perspective.

### 3.3.3. Cyclic Polarization

Cyclic polarization tests were carried out to identify the pitting behavior of the samples. Figure 13 shows the result of the parts with double laser treatment, compared to the untreated samples, where it is demonstrated that no pitting is formed. On the other hand, the vertical part of the straight lines not only indicates that there is no pitting, but also that the material has a passive layer on the surface. This zone is broader in the samples made with the laser remelting strategy.

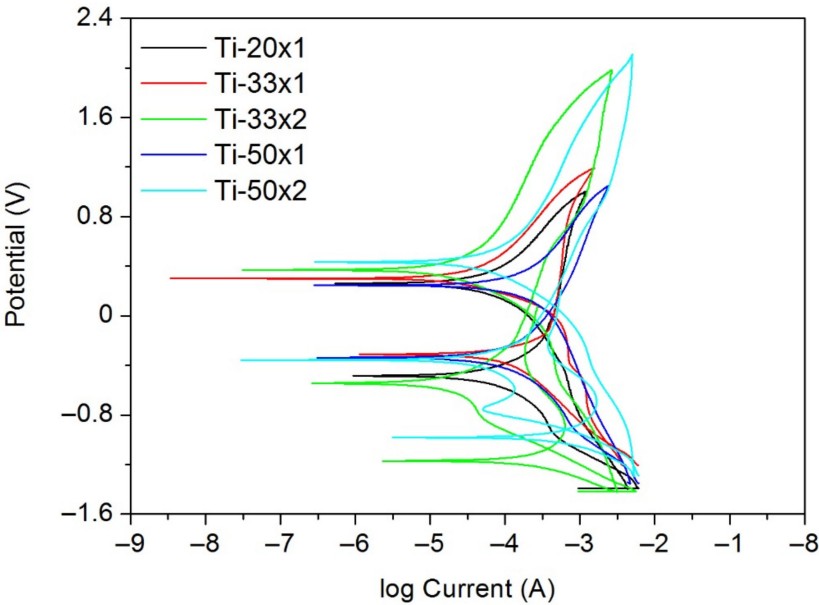

**Figure 13.** Cyclic polarization tests of each specimen after one hour of immersion in 3.5% NaCl solution.

Pitting corrosion nucleation related to passive film breakdown always occurs at defects, where the worst passive film is formed. In the tested samples, although the porosity (together with the lack of fusion) is high, as in the case of Ti-50 × 1, a stable passive layer was formed. The effect of double scanning reduced the number of surface defects, thus reducing the area exposed to the electrolyte, as well as reducing the residual stresses on the surface of the material due to the lower number of metastable phases [41]. In addition, it was seen that the formation of α and β phases promotes the stabilization of the passive layer as well [42]. Since these phases appear when the applied energy is higher, the double scan, just as the low scanning speed, can facilitate the stabilization of the oxide layer.

### 3.3.4. Mass Gain

The mass gain of the tested samples was measured at different immersion times. After one week of immersion in the NaCl solution, no pitting corrosion damage or corrosion products were found on the samples. The mass gain (Figure 14a) was linked to the porosity of the samples. The Ti-50 × 1 sample, which had the biggest porosity, showed the biggest gain in mass (1.2%), and the samples with the lowest porosity, i.e., Ti-33 × 2 and Ti-20 × 1, showed the lowest mass gain (0.41% and 0.19%, respectively). This effect could be explained

by the defect being the parts of the samples that are more prone to corrosion, but no corrosion products were observed at the surface or the defects after the immersion test (Figure 14b), so the value of the gained mass cannot be associated with any corrosion product formed during the exposure time. The simplest consideration is that the presence of porosity favors the accumulation of NaCl that cannot be easily removed with water rinsing.

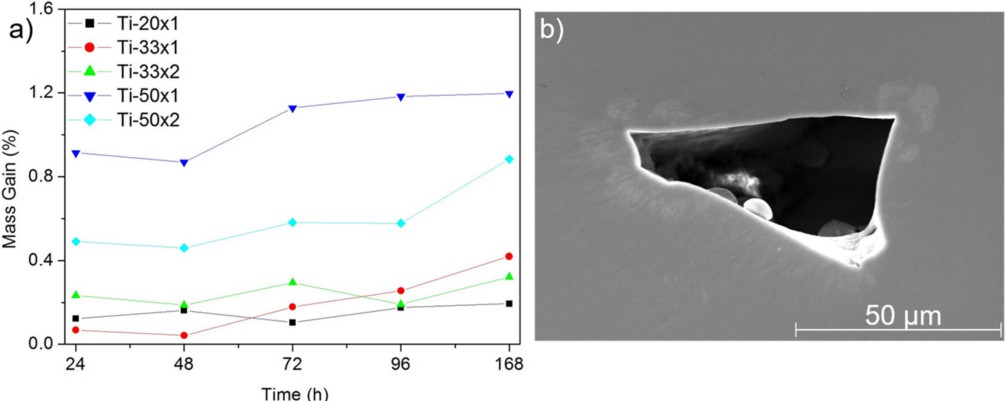

**Figure 14.** (**a**) Evolution of the mass gain of the specimens with immersion time; (**b**) detail of a defect of the material Ti-20 × 1 after 168 h of immersion time.

Therefore, the mass gain of the parts made by SLM with $CO_2$ laser would not be relevant for corrosion processes in chloride media.

The samples manufactured by SLM with a $CO_2$ laser had a relevant number of defects due to the instabilities of the $CO_2$ laser associated with gas being the active media and to the need of directing the laser beam through the air, and to the low absorbance of the laser radiation by the metal powder. Therefore, the samples show more defects than those manufactured with an SLM equipped with a fiber laser. However, it was shown that the presence of these defects does neither affect the hardness of the samples nor the corrosion behavior, which were similar or even better than the parts made by fiber laser had.

The use of a second laser pass could expand the manufacturing window for applying more heat than can be applied in one laser pass without causing the melting of the manufactured structure. In this sense, the use of the laser remelting technique in a 3D printer equipped with a $CO_2$ laser can improve the corrosion behavior of the manufactured parts reducing the number of defects of the specimens and improving its corrosion resistance.

### 4. Conclusions

1. Ti6Al4V specimens were successfully fabricated by L-PBF equipped with a $CO_2$ laser, and a process of laser remelting in each layer was established and was applied to the manufacturing of the samples.
2. The application of a second laser pass reduced the porosity of the samples and caused a slight reduction of the hardness, which was still greater than that of non-AM samples.
3. The grain size of the microstructure of the L-PBF samples increased with the heat input and with the application of a second laser pass. Furthermore, XRD showed that a medium scanning speed and the laser remelting promoted an increment of the β phase due to the transformation of the metastable α′ into α and β.
4. The corrosion behavior of the samples is improved by the application of a second laser pass in each layer, as was revealed by the OCP, *Rp*, and Tafel measurements shown. Furthermore, cyclic polarization tests showed that no pits were generated after one hour of immersion and that the passive layer of the parts fabricated with the laser remelting strategy was more stable than in the single laser pass manufactured samples.

5.  No pitting corrosion damage or corrosion products were found on the surface of the specimens after a week of immersion.

**Author Contributions:** Conceptualization, B.T. and J.R.; methodology, B.T. and J.R.; validation, B.T. and J.R.; formal analysis, J.B., J.d.l.P., B.T. and J.R.; investigation, J.B., J.d.l.P., B.T. and J.R.; resources, A.R., B.T. and J.R.; data curation, J.B. and J.d.l.P.; writing—original draft preparation, J.B., J.d.l.P. and J.R.; writing—review and editing, A.R., B.T. and J.R.; visualization, B.T. and J.R.; supervision, B.T. and J.R.; project administration, A.R., B.T. and J.R.; funding acquisition, A.R., B.T. and J.R. All authors have read and agreed to the published version of the manuscript.

**Funding:** This work was supported by the Ministerio de Economía y Competitividad of Spain (Projects RTI2018-096391-B-C31, PID2021-123891OB-I00, PID2021-124341OB-C21) and Comunidad de Madrid (Projects S2018/NMT-4411 and 2020/00007/019).

**Institutional Review Board Statement:** Not applicable.

**Informed Consent Statement:** Not applicable.

**Data Availability Statement:** Not applicable.

**Conflicts of Interest:** The authors declare no conflict of interest.

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
