# Peer review of "Impact of Remelting in the Microstructure and Corrosion Properties of the Ti6Al4V Fabricated by Selective Laser Melting"

_coatings, doi:10.3390/coatings12020284_

Round 1

Reviewer 1 Report

This paper studied the effects of laser remelting on the microstructures and corrosion properties of Ti6Al4V fabricated by SLM with a CO2 laser. Generally, the paper is well organized and interesting. But some aspects need to be improved before acceptance.

  • There are very few studies on SLM with CO2 laser since most SLM machines are equipped with fiber lasers. I suggest the authors add more relevant references about the application of CO2 SLM and further explain why the authors choose this type of technology.
  • There are some grammatical errors throughout the manuscript. For example, in line 19, the sentence is incomplete. The authors need to carefully check the language.
  • The authors did not mention the layer thickness. This parameter is critical during the manufacturing. I suggest the authors give the parameter in the manuscript.
  • The authors mentioned that the machine is equipped with two CO2 laser of 150 W, but the laser power used for the fabrication of samples is 300 W. This is confusing.
  • Usually, relative density of samples is a critical and essential criterion for the quality evaluation. But the authors did not give these data. In addition, the effects of specific energy input on relative density should be discussed.
  • The differences of microstructures between samples fabricated with and without remelting are not sufficiently investigated. Discussion on the relations between microstructures and corrosion properties should be enhanced.

Author Response

Reviewer #1

This paper studied the effects of laser remelting on the microstructures and corrosion properties of Ti6Al4V fabricated by SLM with a CO2 laser. Generally, the paper is well organized and interesting. But some aspects need to be improved before acceptance.

  • There are very few studies on SLM with CO2 laser since most SLM machines are equipped with fiber lasers. I suggest the authors add more relevant references about the application of CO2 SLM and further explain why the authors choose this type of technology.

We decided to use the CO2 technology to explore if this type of laser could be used for the manufacture of Ti6Al4V samples by SLM. CO2 lasers are much cheaper than other lasers used, such as fiber lasers and they are available in a very wide range of powers, beam qualities, and laser modes. Therefore, the exploration of the limitations and characteristics of CO2 SLM on Ti6Al4V could be very informative for the expansion of limited-cost SLM equipment.

This discussion has been added to the text.

  • There are some grammatical errors throughout the manuscript. For example, in line 19, the sentence is incomplete. The authors need to carefully check the language.

We apologize for the errors and thank the reviewer for the advice. We have revised thoroughly the text to correct the language.

  • The authors did not mention the layer thickness. This parameter is critical during the manufacturing. I suggest the authors give the parameter in the manuscript.

The layer thickness was 60 µm. This parameter is in line 129.

  • The authors mentioned that the machine is equipped with two CO2 laser of 150 W, but the laser power used for the fabrication of samples is 300 W. This is confusing.

The system uses two CO2 lasers with a power of 150 W each. Both lasers travel in parallel and are focussed on the same zone of the sample so that the total power provided is the addition of the laser power of both lasers.

We have included this explanation in the text.

  • Usually, relative density of samples is a critical and essential criterion for the quality evaluation. But the authors did not give these data. In addition, the effects of specific energy input on relative density should be discussed.

We have included the relative density in Figure 6, and we have included the following text on the effect of the specific energy input (Table 1) on relative density.

The increase of the laser input heat from 6.0 W/mm to 9.1 W/mm raised the relative density from 89.7 % to 92.5 % and increasing further the heat input to 15.0 W/mm im-proved further the relative density to 95.3 %. The application of a second laser pass of 6.0 W/mm increased the relative density to 91.8 % and in the case of the use of the sec-ond pass of 9.1 W/mm increased to 94.5 %. Therefore, the higher the heat input, the higher the relative density of the samples. However, an excess in the input heat could melt the samples. Therefore, heat input values above 15 W/mm in the first scan melt the samples, and values above 9.1 W/mm melt the samples in the second scan, with the manufacturing conditions used in this work.

  • The differences of microstructures between samples fabricated with and without remelting are not sufficiently investigated. Discussion on the relations between microstructures and corrosion properties should be enhanced.

We have completed this part of the work with more information on the microstructures of the different samples manufactured. We have added a new figure (the new Figure 9) and the analysis and discussion based on it.

The grain texture was studied by SEM and differences in the grain size and distribution were observed. The grain size of the Ti – 20x2 (Figure 9a) was larger than that of the Ti – 33x1 (Figure 9b) and both greater than that of the Ti – 50x1 (Figure 9c). Therefore, the grain size reduces when increasing the laser scanning speed, i.e., reducing the heat input. The grains found in the Ti – 33x2 (Figure 9d) and Ti – 50x2 (Figure 9e) were greater than the samples with only one laser pass. Therefore, the application of the second scan increased the grains size. Apart from the changes in the number of defects previously described.

We thank the reviewer for the interesting questions that have improved the quality of the work.

Reviewer 2 Report

The article is interesting and essential for those who use Selective Laser Melting (SLM) techniques to produce details with Ti6Al4V alloy. The research aimed to improve the properties of materials produced by SLM with a CO2 laser to reduce the number of defects. The proposed method relies on the double laser scan treatment, i.e. remelting after each layer of the above process. The authors studied the influence of the proposed strategy on the microstructure, defects, hardness, and especially corrosion properties. The application of a second laser pass reduced the porosity of the samples and caused a slight reduction of the hardness. The results obtained indicate that the laser remelting strategy is more stable than the single laser pass manufactured samples.

I have no major substantive comments. From an editorial point of view, I would suggest enlarging the XRD patterns in Fig. 7 for it better readability. 

Author Response

We thank the reviewer for the comments. We have enlarged the XRD patterns in Fig. 7.

Reviewer 3 Report

The proposed manuscript deals with the characterization of some basic properties (microstructure, defects, hardness) of Ti6Al4V alloy printed by selective CO2 laser melting, taking into account the effects due to scan speed and remelting.
In my opinion the main novelty consists in the study of the effect of additional melting on corrosion behavior.

The work is interesting, and seems well articulated, but some issues should be addressed or clarified to better define the value of the work.

1) There is a lack of the characterization of the mechanical property usually investigated first: the tensile behavior. Why were the basic tensile tests not carried out?

2) Throughout the manuscript there is a continuous reference to laser fiber technology. The comparison with similar results found in the literature for fiber laser can be an added value of the proposed work. However, the problem is that it is not specified if these comparisons are made with the same main process parameters. At least every time a datum from the literature is cited to compare it with the results obtained, it would be necessary to specify any differences in the line energy, or in the energy density of the laser: if the differences with those used for the work performed are substantial, the comparisons become not very significant.

3) To better understand the remelting technique, in Section 2.1 the remelting conditions should be spoecified in detail: what is the time interval between first and second melting? does a complete solidification take place between the first and second melting? does the second melting completely melt the layer which is no longer made up of powder but which has become bulk after the first melting?

4) In the analysis of microstructural properties (Section 3.1), the conclusion is: "No differences were observed in the microstructure between the different specimens manufactured and between the samples fabricated with a fiber laser."
The first part of the statement, related to the different specimens manufactured, with particular reference to micrographic analysis, seems rash. In fact, it seems unlikely that the scan speed variation and the remelting will not affect the microstructural morphology, also considering the fact that both the speed variation and the remelting have an effect on the cooling rate, and consequently on the refinement of the microstructure. The fact of presenting the analysis of the micrographs only for one specimen (Ti-33x1, Figure 8) fuels doubts. The question must be clarified, possibly reporting further metallographic investigations for the other specimens.
The second part of the same statement, referred to samples fabricated with fiber laser, should be supported by bibliographical references.

5) At the end of Section 3.2 the decrease in microhardness measured for remelted samples, if compared to corresponding not-remelted ones, is justified by the decrease in the length of the α’ phase, also highlighted in reference [14]. But in the same reference [14] is clearly shown that the hardness increases when one or two remelting are performed (the hardness increases with the number of remelting). This latter result is in contrast with the result reported by the authors, and highlighted by the graph in Figure 9.

6) At lines 362-364 it is stated that "laser remelting has a bigger influence in a range of low scan speeds". The statement is also repeated at lines 377-380: "as it was shown in the previous section, in the range of the lower scan speeds, a laser remelting treatment can result in a better corrosion behavior than a reduction of the scan speed, but this does not occur in the range of higher scan speeds." However, such statements do not seem well supported by the investigation, which in remelted samples was conducted only for two speeds (33 and 50 mm/s). It is therefore not possible to define trends at low and high speeds (it would be necessary to extend the investigation to a greater number of different speeds). It is therefore appropriate to propose more prudent conclusions.

Other minor remarks:
- I suggest to delete the first "laser" in the title, that would become "Impact of remelting in the microstructure and corrosion properties of the Ti6Al4V fabricated by Selective Laser Melting".
- Lines 19-20, the sentence is incomplete.
- Lines 25-27, the sentence repeats what was said just before. Revise the abstract in order to avoid redundant sentences.
- Line 56, it should be "subjected" instead of "submitted".
- Lines 57-59, the sentence is not clear, rephrase it.
- Line 59, what do you mean by the term "hatch"? "hatch spacing" or "offset"? In this case, what stated in the following line 60 "proportion of remolten material, which increases as the hatch also does" is not true (the remelted material reduces as the hatch spacing increases). Clarify this point.
- Line 73, use "difficult" OR "complex", not both togheter.
- Line 83, add "one" between "the" and "reached": "the one reached...".
- Line 88, the statement "these alloys are not used in the aerospace area" is in contrast with what was said in the previous line, and with reality.
- Line 90, the last word should be "field".
- Lines 96-99, the sentence is not clear, rephrase it.
- Line 118, the sentence refers to Figures 2a and 2b (not 1a and 1b). So pay attention to the correct order and number of the figures.
- Line 139, why was a value of 67° chosen for the rotation between one layer and the next?
- Lines 174-182, arrange subscripts in Ecorr and Epic.
- Line 197, delete "o" after "particularly".
- Line 245, add "in" before "SLM".
- Line 318, it should be "decreased" instead of "increased".
- Line 344, at the beginning of the line, delete "and" after "laser".
- Lines 357-360, the sentence is not clear, rephrase it.
- Line 369, the title of Section 3.3.2 would be better as "Tafel tests".
- Lines 438-440, the sentence is not clear, rephrase it.
- Line 449, it should be "XRD" instead of "DRX".
- Lines 452-454, the statement is in contrast with reference [14] (see previous point 5).
- Line 455, add "is" between "samples" and "improved".
- Line 456, the last word should be "shown".
- The types in figures 7, 8, 9, 10, 13 are too small and hard to read, enlarge them.

Author Response

Reviewer #3

The proposed manuscript deals with the characterization of some basic properties (microstructure, defects, hardness) of Ti6Al4V alloy printed by selective CO2 laser melting, taking into account the effects due to scan speed and remelting.

In my opinion the main novelty consists in the study of the effect of additional melting on corrosion behavior.

The work is interesting, and seems well articulated, but some issues should be addressed or clarified to better define the value of the work.

1) There is a lack of the characterization of the mechanical property usually investigated first: the tensile behavior. Why were the basic tensile tests not carried out?

We thank the reviewer for the comment. The influence of remelting in Selective Laser Melting has been investigated in a cited previous study, ref [14], but the influence of the remelting in the corrosion behavior on parts made by Selective Laser Melting has not been studied. Due to this, we have focused on studying the corrosion behavior and the electrochemical study. However, following the reviewer’s comments, we will work in another article to evaluate the mechanical properties of Ti6Al4V manufactured by L-PBF using a CO2 laser.

2) Throughout the manuscript there is a continuous reference to laser fiber technology. The comparison with similar results found in the literature for fiber laser can be an added value of the proposed work. However, the problem is that it is not specified if these comparisons are made with the same main process parameters. At least every time a datum from the literature is cited to compare it with the results obtained, it would be necessary to specify any differences in the line energy, or in the energy density of the laser: if the differences with those used for the work performed are substantial, the comparisons become not very significant.

We thank the reviewer for the comment. We focused the comparison on what other authors considered their optimum outcome. Following the comment, we have included more information on the processing parameters that other authors have used, and that improve the understanding of the results we have obtained.

3) To better understand the remelting technique, in Section 2.1 the remelting conditions should be specified in detail: what is the time interval between first and second melting? does a complete solidification take place between the first and second melting? does the second melting completely melt the layer which is no longer made up of powder, but which has become bulk after the first melting?

Following the reviewer's comment, we have completed the remelting technique in the experimental section.

The samples were manufactured using 300 W of laser power, using the combined power of two lasers of 150 W that were focused in the same point; and the scanning speeds used were 50, 33, and 20 mm/s. A set of samples was manufactured using the standard methodology of melting the metallic powder after its deposition with the scanning speeds indicated.

A second set was manufactured using a double laser scanning method. Firstly, a layer of powder is spread on the surface. Then, the laser scans the programmed zones and there melts the powder. Afterward, after finishing the first scan in all the programmed zones, the laser repeats the previous pattern and applies a second melting process without adding more powder. From the first scan to the second one, the elapsed time was 10 s. Then, the next powder layer was deposited, and a 67° rotation angle between the laser scanning was applied. Table 1 resumes the different types of samples manufactured. The samples with a scan speed of 20 mm/s could not be made manufactured with a second set due to excessive heat, which caused its partial melting.

4) In the analysis of microstructural properties (Section 3.1), the conclusion is: "No differences were observed in the microstructure between the different specimens manufactured and between the samples fabricated with a fiber laser."

The first part of the statement, related to the different specimens manufactured, with particular reference to micrographic analysis, seems rash. In fact, it seems unlikely that the scan speed variation and the remelting will not affect the microstructural morphology, also considering the fact that both the speed variation and the remelting have an effect on the cooling rate, and consequently on the refinement of the microstructure. The fact of presenting the analysis of the micrographs only for one specimen (Ti-33x1, Figure 8) fuels doubts. The question must be clarified, possibly reporting further metallographic investigations for the other specimens.

The second part of the same statement, referred to samples fabricated with fiber laser, should be supported by bibliographical references.

The reviewer is right and we apologize for the deficiencies in our explanation. We were referring to the martensitic structure of the Ti6Al4V and we did not study the texture of the manufactured samples.

Following the suggestion, we have completed the analysis and included a new image. The new information is the following.

The grain texture was studied by SEM and differences in the grain size and distribution were observed. The grain size of the Ti – 20x2 (Figure 9a) was larger than that of the Ti – 33x1 (Figure 9b) and both greater than that of the Ti – 50x1 (Figure 9c). Therefore, the grain size reduces when increasing the laser scanning speed, i.e., reducing the heat input. The grains found in the Ti – 33x2 (Figure 9d) and Ti – 50x2 (Figure 9e) were greater than the samples with only one laser pass. Therefore, the application of the second scan increased the grains size. Apart from the changes in the number of defects previously described.

Figure 9. Microstructure of the different specimens tested as observed by SEM: a) Ti – 20x1; b) Ti – 33x1; c) Ti – 33x2; d) Ti – 50x1; and e) Ti – 50x2.

Also, other parts of the revised manuscript take advantage of this added information.

5) At the end of Section 3.2 the decrease in microhardness measured for remelted samples, if compared to corresponding not-remelted ones, is justified by the decrease in the length of the α’ phase, also highlighted in reference [14]. But in the same reference [14] is clearly shown that the hardness increases when one or two remelting are performed (the hardness increases with the number of remelting). This latter result is in contrast with the result reported by the authors, and highlighted by the graph in Figure 9.

We thank the reviewer for this comment. We apologize for the confusion in this case. There are differences between the results of the cited articles and ours. We have modified the text to explain the differences.

In all cases, the application of a second laser scan caused a slight reduction in the microhardness of the samples. The remelting process has two opposite effects on hardness [35]. On the one hand, we observed that the application of multiple remelting and reheating processes promoted the growth of the grain size, and possible the transformation of the α’ phase into the α and β phases. Both effects could cause a reduction of the microhardness of the samples. On the other hand, the increase in the relative density of the samples could increase hardness and thermal conductivity. For the conditions tested in this study, the result of these changes was a slight reduction in hardness. However, another study indicated that hardness increased with the application of one or two remelting processes using a fiber laser [14]. The main difference between the works is the input heat, which was 0.6 W/mm in their case and higher than 6 W/mm in ours. The order of magnitude in difference indicates that, at all conditions, the remelting we caused was much greater and, hence, the cooling rate was slower. They showed this phenomenon by a reduction in the lattice parameters, although they did not study the grain size.

6) At lines 362-364 it is stated that "laser remelting has a bigger influence in a range of low scan speeds". The statement is also repeated at lines 377-380: "as it was shown in the previous section, in the range of the lower scan speeds, a laser remelting treatment can result in a better corrosion behavior than a reduction of the scan speed, but this does not occur in the range of higher scan speeds". However, such statements do not seem well supported by the investigation, which in remelted samples was conducted only for two speeds (33 and 50 mm/s). It is therefore not possible to define trends at low and high speeds (it would be necessary to extend the investigation to a greater number of different speeds). It is therefore appropriate to propose more prudent conclusions.

We appreciate the comment of the reviewer. Previously to start making specimens for this work, we did a parameter study to optimize some parameters, like the scan speed. In this way, we found that from the 33 mm/s scan speed the defects were minimized, and no significant differences were found in the porosity at lower scan speeds. Once we started fabricating the specimens for this study, we made parts at the optimized scan speed, 33 mm/s, one at a lower speed (20 mm/s) and one at a higher scan speed (50 mm/s) and we executed the remelting in every part. However, the specimens made at 20 mm/s could not be fabricated with the remelting due to the excessive input heat, which resulted in the deformation of the parts.

On the other hand, these differences in the scan speeds are very influential on the final parts, so the working range for this study is wide. Nevertheless, we recognize that we must stick to the studied range of parameters, and we have modified the text to focus on the conclusions that are supported by our results.

Other minor remarks:

- I suggest to delete the first "laser" in the title, that would become "Impact of remelting in the microstructure and corrosion properties of the Ti6Al4V fabricated by Selective Laser Melting".                                                             Done

- Lines 19-20, the sentence is incomplete.                                                                                                       Done

- Lines 25-27, the sentence repeats what was said just before. Revise the abstract in order to avoid redundant sentences.                                                                                                                                                                                    Done

- Line 56, it should be "subjected" instead of "submitted".                                                                         Done

- Lines 57-59, the sentence is not clear, rephrase it.                                                                                      Done

- Line 59, what do you mean by the term "hatch"? "hatch spacing" or "offset"? In this case, what stated in the following line 60 "proportion of remolten material, which increases as the hatch also does" is not true (the remelted material reduces as the hatch spacing increases). Clarify this point.                                                                                                                Done

- Line 73, use "difficult" OR "complex", not both togheter.                                                                         Done

- Line 83, add "one" between "the" and "reached": "the one reached...".                                              Done

- Line 88, the statement "these alloys are not used in the aerospace area" is in contrast with what was said in the previous line, and with reality.                                                                                                                                                     Done

- Line 90, the last word should be "field".                                                                                                        Done

- Lines 96-99, the sentence is not clear, rephrase it.                                                                                      Done

- Line 118, the sentence refers to Figures 2a and 2b (not 1a and 1b). So pay attention to the correct order and number of the figures.                                                                                                                                                                      Done

- Lines 174-182, arrange subscripts in Ecorr and Epic.                                                                                   Done

- Line 197, delete "o" after "particularly".                                                                                                       Done

- Line 245, add "in" before "SLM".                                                                                                                     Done

- Line 318, it should be "decreased" instead of "increased".                                                                       Done

- Line 344, at the beginning of the line, delete "and" after "laser".                                                           Done

- Lines 357-360, the sentence is not clear, rephrase it.                                                                                 Done

- Line 369, the title of Section 3.3.2 would be better as "Tafel tests".                                                       Done

- Lines 438-440, the sentence is not clear, rephrase it.                                                                                 Done

- Line 449, it should be "XRD" instead of "DRX".                                                                                            Done

- Lines 452-454, the statement is in contrast with reference [14] (see previous point 5).                    Done

- Line 455, add "is" between "samples" and "improved".                                                                            Done

- Line 456, the last word should be "shown".                                                                                                  Done

- The types in figures 7, 8, 9, 10, 13 are too small and hard to read, enlarge them.                               Done

- Line 139, why was a value of 67° chosen for the rotation between one layer and the next?

The 67° angle was chosen as it is the one used in many industrial L-PBF systems because it avoids the accumulation of parallel scans between closer layers, so, in theory, it derives in more isotropic manufacturing in the basal plane of the build-up.

This explanation has been included in the text.

We have followed all the suggestions made by the reviewer and we appreciate the detail of the analysis made as they have helped to improve the work.

Round 2

Reviewer 1 Report

The paper can be accepted in its present form.

Author Response

We thank the reviewer for the consideration of accepting our work.

Reviewer 3 Report

The manuscript has been improved. The authors have substantially addressed the comments from the original review. The revised manuscript can be accepted for publication.

Just some last remarks:

1) Lines 58-60, the hatch distance should be the spacing between two successive adjacent passes of the laser beam. If so, the remelted material reduces as the hatch distance increases. Instead you state: "...the proportion of remolten material, which increases as the hatch distance also does...". Clarify this point.

2) Lines 127-131, provide references for 67° rotation angle between one layer and the next.

3) Line 189, arrange subscripts in Ecorr.

Author Response

We appreciate the comments and detail of the revision of the work made by the reviewer:

Just some last remarks:

1) Lines 58-60, the hatch distance should be the spacing between two successive adjacent passes of the laser beam. If so, the remelted material reduces as the hatch distance increases. Instead you state: "...the proportion of remolten material, which increases as the hatch distance also does...". Clarify this point.

The reviewer is right, and we made an error. The correct expression is that "the proportion of remolten material, which decreases as the hatch distance also does...

2) Lines 127-131, provide references for 67° rotation angle between one layer and the next.

We have added two references that show that this procedure is commonly used in SLM.

3) Line 189, arrange subscripts in Ecorr.

We have corrected the subsctripts.